# Impact of Home-Based Work during the COVID-19 Pandemic on Mental and Physical Health in a German Population-Based Sample

**DOI:** 10.3390/healthcare12070789

**Published:** 2024-04-05

**Authors:** Franziska Emmerich, Julia Junghans, Markus Zenger, Elmar Brähler, Yve Stöbel-Richter, Lisa Irmscher, Ernst Peter Richter, Hendrik Berth

**Affiliations:** 1Division of Psychosocial Medicine and Developmental Neurosciences, Research Group Applied Medical Psychology and Medical Sociology, Faculty of Medicine Carl Gustav Carus, Dresden University of Technology, 01307 Dresden, Germany; 2Department of Differential and Personality Psychology, University of Applied Sciences Magdeburg-Stendal, 39114 Stendal, Germany; 3Clinic and Polyclinic for Psychosomatic Medicine and Psychotherapy/Behavioral Medicine, University of Leipzig Medical Center, 04103 Leipzig, Germany; 4Department of Psychosomatic Medicine and Psychotherapy, University Medical Center Mainz, 55131 Mainz, Germany; elmar.braehler@medizin.uni-leipzig.de; 5Department of Medical Psychology and Medical Sociology, University of Leipzig Medical Center, 04103 Leipzig, Germany; 6Faculty of Managerial and Cultural Studies, The University of Applied Sciences Zittau/Görlitz, 02763 Zittau, Germany

**Keywords:** COVID-19 pandemic, home-based-work, working from home, mental health, psychological strain, physical complaints, Saxon Longitudinal

## Abstract

*Background*: The COVID-19 outbreak necessitated physical distancing, as part of secondary prevention, at a personal and professional level. Working from home (WFH) became increasingly important. In this study, the impact of the COVID-19 pandemic restrictions on physical and mental health is investigated, compared with pre-pandemic data, and with employees who WFH and are on-site. *Methods*: Data from the German Saxon longitudinal study population were used. Attitudes towards WFH as well as mental and physical health assessments during the COVID-19 pandemic were examined. Comparisons were made with corresponding pre-pandemic scores and between employees WFH and on-site in 2022. *Results*: In total, 319 participants with equal gender distribution were included. Of those, 86 worked from home stating better organizability of their work, more time for partnership, less stress, and greater work satisfaction. Compared to pre-pandemic data, the D-score, PHQ-4, G-Score, and PHQ-SSS-8 showed a significant increase. No difference in physical or mental health between employees WFH and on-site was observed. *Conclusion*: In general, COVID-19 restrictions had a negative impact on mental and physical health. Although WFH is well accepted, it did not show significant health benefits.

## 1. Introduction

The SARS-CoV-2 (Severe Acute Respiratory Syndrome Coronavirus-2) caused one of the most devastating pandemics of the 21st century [1]. More than 270 million people in Europe and Central Asia have been infected with COVID-19, and more than 2.2 million people have died from the disease since January 2020 [2]. Due to the increasing number of infections and the rapid spread of COVID-19, the WHO declared an international health emergency on 30 January 2020 [3].

In Germany, the COVID-19 pandemic was characterized by fierce waves of infection and disease from 2020 to 2022 [4]. To contain the pandemic, the German government decided on restrictions on public life. Cultural offerings were reduced or discontinued. Theatres stopped playing and the German Football League was suspended (12/13 March 2020). Controls and entry bans at German borders were carried out and schools as well as kindergartens were closed (1 March 2020). The restrictions on public life were tightened. Gatherings of more than two people were banned and restaurants as well as hairdressers and stores had to close (20 March 2020) [5].

Several “lockdowns” and associated restrictions were aimed at maintaining sufficient capacity to treat COVID-19 patients and investigate suspected cases while ensuring infection protection for the population. The public was called upon to minimize their risk of infection by following the “AHA rules” (distance, hygiene, everyday masks) and thus containing the spread of the virus [6]. In the resolution “Einschränkungen des öffentlichen Lebens zur Eindämmung der COVID-19-Epidemie” adopted by the German Chancellor and the heads of government of the German states, social distancing was considered as the best way to prevent illness from COVID-19 [7].

The authors of the review by Röhr et al. identified a variety of psychological reactions in connection with quarantine measures, including depressiveness, anxiety, anger, stress, post-traumatic stress, social isolation, loneliness, and stigmatization [8]. Other international studies have also found an increase in psychological strain, in particular increased depressive symptoms, more severe stress, and anxiety disorders during the coronavirus pandemic [9,10,11].

Social contact should not only be minimized in private life, the government also appealed to companies to take advantage of home working options wherever feasible [7]. As a result, an increasing willingness on the part of companies to give their employees the opportunity of working from home (WFH) was observed. In 2019, the year before the pandemic, the percentage of workers WFH was 12.8%, while this percentage increased in subsequent pandemic years (2020 = 21.0%, 2021 = 25%, 2022 = 24%) [12]. WFH was not only beneficial as a prevention measure but also allowed employees flexible working hours, less stress, more perceived autonomy as well as self-responsibility, and greater job satisfaction [13,14,15]. However, a look at the current literature of international surveys shows advantages and disadvantages as described below. On the other hand, with increasing duration and intensity of home-based work, negative factors such as social isolation, work–life conflicts, less productivity, stress, and sleep problems, while WFH [13,16,17], as well as having insufficient ergonomic workplace conditions, might lead to health issues [18,19]. In addition, Xiao et al. reported on factors that affect not only the health but also the behavior of employees, and thus may have had a direct impact on both physical and mental well-being during the pandemic (less physical activity, unhealthy diet, additional childcare at home, distractions at work, less communication with colleagues, or changed working hours) [20]. Therefore, adverse effects can affect not only the psychological but also the physical well-being of employees. To further address this issue, a cohort is needed from which data on mental and physical health were collected in the years before the pandemic and the years following. German longitudinal studies, which investigated the impact of the COVID-19 pandemic on physical and mental health, began their study period in 2020 [21,22]. To our knowledge, there are no longitudinal studies in Germany that have examined a longer comparison period before and during the pandemic, and only one study examined participants from 2019 to 2020 [23]. In our study, we comprise data from 2017 to 2022 allowing us not only to compare pre-pandemic and pandemic data but also possible changes during COVID-19.

We investigated the research questions of how the COVID-19 pandemic affected mental and physical strain. For this purpose, we used data from the Saxon longitudinal study, which have been collected almost continuously since 1987. This gave us the unique opportunity to compare data collected years before the pandemic with data collected at the beginning and during the pandemic. Moreover, the characteristic of WFH was integrated into the survey for the first time in 2022 due to its increasing relevance and was analyzed in light of mental and physical stress.

## 2. Materials and Methods

### 2.1. Sample

The data for this study were taken from the Saxon Longitudinal Study (SLS, “Sächsische Längsschnittstudie”) which was conducted for the first time in 1987 in Leipzig and Chemnitz [24]. The study concept arose from scientific research on youth development in the German Democratic Republic (GDR) at the Central Institute for Youth Research. The survey of the first wave was carried out in cooperation with the Karl Marx University of Leipzig and the Zwickau University of Education and represented a special data collection for the time and the geographical limitations.

In this study, data from wave W30 (survey year 2017/18, *N* = 314), W31 (survey year 2019/20, *N* = 323), W32 (survey year 2021, *N* = 321), and W33 (survey year 2022, *N* = 319) were used. The data of the 33rd wave was collected from 1 September 2022 to 31 December 2022.

The initial cohort consisted of 14-year-old east Germans who were in their 8th year of school (*N* = 1407). In total, pupils from 72 different classes and 41 various schools across the districts of Karl–Marx–Stadt and Leipzig were randomly selected. Both genders were almost equally distributed (female 47.2%). After the third wave (year 1989), 45.82% (*N* = 587) of 1281 respondents agreed to continue participating in the survey. In the following waves, more than 300 participants were recorded almost every year. Since 2002, the survey has been supplemented by various validated and standardized questionnaires to assess subjective physical and psychological health. In addition, other topics such as unemployment experience, starting a family, COVID-19, or WFH were included. Since 2010, it has been possible to complete the SLS questionnaire online, in addition to the paper version [24]. For completing the questionnaire, participants were rewarded with a financial contribution of 25€ (W30), 25€ (W31), 30€ (W32), and 35€ (W33).

The survey was conducted with the approval of the responsible ethics committee (Ethics Committee of the Technische Universität Dresden, EC8012011, approval date: 11 February 2011), in accordance with national law and in accordance with the Declaration of Helsinki of 1975 (in the current, revised version).

### 2.2. Sample Description

In total, we collected data from more than 300 participants in each wave (wave 30 (2017) = 314; wave 31 (2019) = 323; wave 32 (2021) = 321; wave 33 (2022) = 319). Regarding gender and age, the respondents were almost evenly distributed (average year of birth was 1973). At the time of the survey (2022), the median age was 50 years and the majority of respondents lived in a partnership (>80%). Moreover, the proportion of households in which one child lived was 41.7% (see Table 1).

### 2.3. Measurements

In addition to general data, various instruments were used to assess mental and physical health as well as attitudes towards WFH.

#### 2.3.1. Psychological Strain

Psychological exposure was measured using the D-Score, the PHQ-4 (Patient Health Questionnaire-4), the LS-S (Loneliness Scale), the L-1 (short scale of life satisfaction), and a Corona Anxiety scale.

The D-score was developed especially for the SLS [25]. It measures the psychological stress of the participants based on the items dejection, meaninglessness of life, hopelessness, and fear of the future. Points were assigned to the answer options, which were added up. This value represents the D-score and can range from 0 (no stress) to 8 (high stress) [25]. The internal consistency of the D-Score for wave 33 was calculated using Cronbach’s Alpha α = 0.826 and McDonald’s Omega Ω = 0.891 (W30 α = 0.812, Ω = 0.813; W31 α = 0.824, Ω = 0.833; W32 α = 0.838, Ω = 0.849).

PHQ-4 (Patient Health Questionnaire-4) is a short questionnaire consisting of a two-item depression scale (Patient Health Questionnaire-2, PHQ-2) and a two-item anxiety scale (Generalized Anxiety Disorder, GAD-2) [26]. The PHQ-2 measures the two cardinal symptoms of depression. Respondents indicate on a 5-point Likert scale how often they have felt affected by “little interest or pleasure in your activities” and “dejection, melancholy, or hopelessness” within the past two weeks [27]. GAD-2 is a module to assess generalized anxiety disorder [28]. Participants indicate on a 5-point Likert scale how often they felt affected by the complaints of “nervousness, anxiousness or tension” and “unable to stop or control worry” within the last two weeks. These four questions form a score, which ranges between 0 and 12 points, with a higher value being associated with a higher degree of depression or anxiety. The internal consistency of the PHQ-4 for wave 33 was assessed using Cronbach’s Alpha α = 0.928 and McDonald’s Omega Ω = 0.930 (W30 α = 0.853, Ω= 0.854; W31 α = 0.857, Ω = 0.857).

The LS-S (Loneliness Scale) includes three items that address the feeling of being left out, the lack of sociability, and the feeling of being socially isolated. Respondents indicate how often such feelings occur from “very often” to “never” [29] on a 5-point Likert scale. The sum score ranges between 3 and 15 points. To assess the internal consistency of the score for wave 33, Cronbach’s Alpha (α = 0.811) and McDonald’s Omega (Ω = 0.792) were calculated (W30 α = 0.811, Ω = 0.815).

The L-1 (short scale of life satisfaction) consists of one item that records life satisfaction. This measurement instrument was taken from the Socio-Economic Panel (SOEP). The formulation is “All in all, how satisfied are you with your present life?” which can be classified by the respondents on an 11-point scale. It ranges from (0) “not at all satisfied” to (10) “completely satisfied” [30].

The Corona Anxiety Scale is an instrument to assess anxiety in the context of the COVID-19 pandemic. The scale consists of seven individual questions and was developed for the SLS and evaluated for the first time in wave 32 (2021) of the SLS. The questions of this measuring instrument address aspects such as fear of infection, uncontrolled infection and spread of the virus, serious consequences of infection, and worries. The questions were answered from “strongly disagree” to “strongly agree” on a 5-point Likert scale. The point totals were summed up. The score ranges between 5 and 35 points, whereby a higher value corresponds to a greater corona-related fear [31]. Cronbach’s Alpha of α = 0.791 as well as McDonald’s Omega of Ω = 0.751 were assessed to address the internal consistency of the score (W32 α = 0.814, Ω = 0.818).

#### 2.3.2. Physical Strain

The G-score, SSS-8 (Somatic Symptom Scale), and the individual item on health status were used to assess physical stress.

The G-Score was developed for the SLS as a screening instrument for the self-assessment of physical health. The occurrence of the four symptoms nervousness, stomach complaints, sleeping problems, and heart complaints within the last 12 months was queried. The assessment was completed on a four-point Likert scale (“yes, often” (3), “yes, once in a while” (2), “yes, rarely” (1), “no, never” (0)). The points were added up, resulting in a scale ranging between 0 and 12 points. The higher the summed score, the more frequently physical complaints occur [32]. For wave 33 (2022), the internal consistency of the score was calculated using Cronbach’s Alpha α = 0.758 and McDonald’s Omega Ω = 0.787 (W30 α = 0.728, Ω= 0.734; W31 α = 0.762, Ω = 0.765; W32 α = 0.760, Ω = 0.768).

The SSS-8 is a shortened 8-item version of the PHQ-15 questionnaire. The scale is a screening instrument that assesses the presence and severity of common somatic symptoms like abdominal or digestive discomfort, back pain, and r sleep disturbances. Participants rated the occurrence of these symptoms within the last 7 days on a five-point Likert scale [33]. Cronbach’s Alpha of α = 0.856 and a value for McDonald’s Omega of Ω = 0.835 were calculated (W30 α = 0.831, Ω = 0.830).

The single-item health status is known as a health indicator for the subjective assessment of one’s state of health. The question, “How would you assess your current state of health?”, showed to be a reliable predictor of mortality trends [34]. There were five response options to choose from: “very good”, “good”, “satisfactory”, “less good”, and “poor”.

#### 2.3.3. Attitudes towards WFH

To investigate attitudes towards WFH, eight questions were integrated into the SLS for wave 33 (2022). These items were selected from the “Home Office Barometer 2020” developed by the research institute gsf.bern AG [35]. Four items cover the sub-topics loneliness, stress, childcare, and future home-based work. The participants answered the questions on a 5-point Likert scale with the response options “fully agree” to “not at all” (1 to 5 points). Additionally, three questions were included dealing with personal attitudes comprising the following topics: productivity, partnership, and childcare. For those three questions, a 5-point Likert scale was applied with the options “strongly agree” to “strongly disagree” (1 to 5 points). Finally, satisfaction with WFH was assessed using the question: “How satisfied are you with your personal WFH situation?”, which could be answered with “very satisfied”, “rather satisfied”, “not assessable”, “rather dissatisfied”, or “very dissatisfied” (1 to 5 points). Results were evaluated not only for all participants in the W33 but also for those respondents WFH during the survey period and those who did not.

### 2.4. Statistical Analysis

A descriptive analysis was carried out for socio-demographic characteristics and for the scales used (G-score, D-score, etc.) as well as for the questions on WFH. We calculated Wilcoxon tests for group comparisons of the psychological and physical strain scores between the survey years W30 (2017)–W33 (2022). In addition, differences in attitudes towards WFH were calculated using Chi^2^-tests. Analyzed groups included the whole cohort, WFH participants and not WFH participants. For all test procedures, a significance level of 𝛼 = 0.05 was chosen. The sample size calculation was completed by using G*Power 3.1.3 (Axel Buchner, Edgar Erdfelder, Franz Faul, Albert-Georg Lang, Heinrich Heine University, Düsseldorf, Germany), applying an effect size of Cohen’s d = 0.5, and a power of 95% (1 − β = 0.95) resulting in a sample of at least *N* = 184 [36]. A test of normality was conducted using the Kolmogorv–Smirnov test. If variables followed a normal distribution, means and standard deviations were used, but otherwise median and interquartile ranges were used.

To test for the internal consistency of the measurement instruments, Cronbach’s Alpha and McDonald’s Omega were calculated. Statistical analysis was carried out with IBM SPSS Statistics (version 28, International Business Machines Corporation (IBM), Armonk, NY, USA).

## 3. Results

### 3.1. Psychological Strain

The level of psychological stress was obtained using the D-Score, which was found to be the lowest in wave 30 (2017). Significant average differences were found between the waves 31 (2019) and 32 (2021) as well as the waves 32 (2021) and 33 (2022) (*p* < 0.001). Meaning, there was a difference between the mean D-scores prior to the COVID-19 pandemic (2019) and during the pandemic (2022), as well as between the pandemic years 2021 and 2022 (see Table 2).

Depressiveness represented by the PHQ-2, revealed the lowest mean values in wave 30 (2017). Before the pandemic, we determined a value of (2.85 (1.15)), which increased to (3.55 (1.96)) by 2022. A Wilcoxon test calculation revealed significant changes between waves 30 (2017) and 33 (2022) as well as between wave 31 (2019) and 33, *p* < 0.001 (see Table 2).

Anxiety, measured with the help of the GAD-2-Score was found to be lowest in wave (2019) (2.92 (1.18)), shortly before the COVID-19 outbreak. Throughout the pandemic, significant differences between waves 30 (2017) and 33 (2022) (*p* < 0.01) and between waves 31 (2019) and 33 (2022) (*p* < 0.01) were obtained (see Table 2).

We found the lowest mean value for PHQ-4 in wave 31 (2019) (5.74 (2.16)), which rose to (7.14 (3.86)) by 2022 (wave 33). Significant differences were observed between the pre-pandemic wave 30 (2017) and wave 33 (2022) (*p* < 0.01) as well as between pre-pandemic wave 31 (219) and 33 (2022) (*p* < 0.01) (see Table 2).

With reference to loneliness measured using the LS-S, the lowest mean value was found in W30 (2017). A slightly higher value was obtained in wave 33 (2022). Overall, no significant difference was found between the waves before (wave 30 (2017)) and during the COVID-19 pandemic (wave 33 (2022)) (*p* > 0.05) (see Table 2).

Addressing the satisfaction of life with the L-1-Score, the highest value was recorded in wave 30 (2017). In wave 31 (2019) and 32 (2021), the mean values of life satisfaction continuously decreased and slightly increased in wave 33 (2022). Altogether, no significant difference was found between the years before and during the COVID-19 pandemic (*p* > 0.05) (see Table 2).

Pandemic anxiety was determined using the Corona Anxiety Scale for the years 2021 and 2022, with a statistical difference being identified (*p* < 0.01) (see Table 2).

### 3.2. Physical Strain

To measure the level of physical health, the G-Score was calculated for the waves 30–33 (2017–2022), as described in Table 2. The lowest G-score was calculated in wave 32 (2021) (3.58 (2.92)). Before the pandemic, a score of 3.77 (2.89) was determined in wave W31 (2019), which fell in the following year but rose again in wave 33 (3.76 (2.88)). A significant difference was found between wave 31 (2019) and wave 32 (2021) (*p* = 0.038).

Somatic symptoms were measured with the help of the SSS-8 score which was found to be the highest in the year of the pandemic (wave 33, 2022). Pre-pandemic, a lower score was detected in the year 2017 (wave 30). The comparison of wave 30 (2017) and 33 (2022) revealed a significant difference (*p* < 0.001) (see Table 2).

Addressing the personal thoughts on the level of the individual health status, the lowest value and, therefore, the best participant-estimated health state was found prior to the pandemic in wave 30 (2017). Values of wave 31 (2019) and 32 (2022) showed comparable results. However, an increasing tendency was observed in wave 33 (2022) even though no significant differences were detected between the years (*p* > 0.05) (see Table 2).

### 3.3. Attitudes towards WFH

When looking at the group comparisons (WFH vs. not WFH participants), we found significant differences in attitudes towards WFH on the following topics: feelings of being alone and reduction in stress when WFH, productivity when WFH, control of working hours when WFH, compatibility with partnership, satisfaction with home-based work, and future WFH after the pandemic. No significant difference was found in the question of the compatibility of WFH with childcare (see Table 3).

Looking at the absolute and relative figures, we found a more positive attitude among WFH respondents in terms of feelings of being alone, less stress at work, the compatibility of WFH with partnership, more productivity, and control over working hours. In addition, the majority of WFH respondents were satisfied with their individual working situation at home, and over 80% stated that they would like to continue WFH after the pandemic. Detailed information can be found in Table 3.

### 3.4. Influence of Working from Home on the Psychological and Physical Strain on Employees

To investigate the influence of WFH on psychological or physical strain, we used non-parametric tests (Wilcoxon tests) to calculate whether there were significant differences between the two independent samples of “employees who WFH” and “employees who did not WFH “. We used the previously presented psychological and physical health measurement scores for the analysis. As presented in Table 4, we did not observe a significant difference between the two sample groups in any of the scores collected neither for psychological nor for physical health.

## 4. Discussion

In this study, we investigated the impact of the COVID-19 restrictions on mental well-being and physical health by using selected scores: D-Score, PHQ-4, LS-S, L-1, G-Score, SSS-8, and the single-item health status. We assessed changes before and during the pandemic and compared employees who worked from home with those who worked on-site. Moreover, we determined a descriptive overview of attitudes towards WFH.

### 4.1. Psychological Strain

Our results on psychical strain indicate a potential negative impact of the COVID-19 restrictions on mental health. We indicated an increase in depressive symptoms throughout the pandemic as well as when comparing data before and during the pandemic. These findings were in line with other studies [37,38,39,40]. A possible explanation for this increase may have been rising mental strain during the pandemic. This is also reflected by an increase in anxiety symptoms and stress experiences, which was also depicted in our sample when comparing the data before and during the coronavirus pandemic. The pandemic was an exceptional situation posing many challenges. Restrictions in personal and professional life and the fear of falling ill or worries about relatives are just a few examples of this demanding time. Our findings on the longitudinal trends in depressive, anxiety, and stress symptoms are in line with those reported in the current literature [37,39,41,42], e.g., the authors of the Swiss study conducted by Piumatti et al. observed an increase in the prevalence of depression (moderate to severe) from 7.5% to 12.5%, anxiety from 4.8% to 8.1%, and stress from 5.5% to 8.8% (from August 2020 to May 2021) [41]. Nevertheless, this increase in psychological stress during the pandemic is discussed as a reaction to the exceptional situation of the global coronavirus pandemic [43].

However, the decrease in corona-associated anxiety observed and the lack of significant changes in loneliness contrast with other studies [43,44]. Nevertheless, Landmann and Rohmann found that physical loneliness increased during the pandemic, while emotional and social loneliness remained stable, which is consistent with our findings [45]. Moreover, studies conducted later in the pandemic reported similar results [46]. These temporal trends may indicate that the adaptation to the new, unfamiliar pandemic situation initially leads to an observable increase in corona-related anxiety and loneliness, and then normalizes again after habituation and may lead to the development of coping mechanisms. Therefore, the period of our survey may influenced the results obtained, as the pandemic was already on the retreat in Germany from September to December 2022 [47].

In addition, we observed stable life satisfaction in our study during the coronavirus pandemic which is consistent with the current literature [48,49]. Possibly, life satisfaction remains stable even in times of crisis if the individual adjusts their standards, e.g., adjusting temporal, social, or health comparisons downwards [50,51].

### 4.2. Physical Strain

Regarding physical complaints, we observed significant increases in the G-score and SSS-8 when comparing data before and during the pandemic. Consistent with our findings on the G-score, Refle et al. found little immediate impact of the crisis on physical health in their study, conducted from March to June 2020 [52]. This was also supported by other studies reporting minor or no differences in the occurrence of physical symptoms compared with the pre-pandemic levels [53,54]. Minor differences were, e.g., found in the study conducted by Waltersbacher et al. who observed an increase in emotional upsets and a slight increase in psychosomatic complaints [54]. The authors attribute this variation in physical well-being to individual changes in the work situation and changes in private circumstances such as additional childcare or personal resilience [54].

Our survey showed an increase in the Somatic Symptom Scale, which is in line with the current literature [55,56,57]. In the study conducted by Söğütlü and Göktaş, a significant increase in somatic symptoms was observed, especially in participants with a history of COVID-19 contact, hospitalizations due to COVID-19, and hospitalizations of close relatives with a positive test result [56]. High levels of health anxiety may be related to serious misinterpretations of physical sensations and changes, dysfunctional beliefs about one’s health or illness, and poorly adapted coping behaviors [58]. Moreover, dizziness and chest pain, in particular, may be long-term consequences of having undergone COVID-19 infection, and thus some of the respondents with somatic complaints may be suffering from long-COVID syndrome [57]. Overall, the extraordinary pandemic situation, which led to changes in the workplace such as working from home, less contact with colleagues, additional stress due to family obligations and health anxiety, may have had an impact on physical well-being and led to more somatic complaints.

Paradoxically, we did not observe a decline in the assessment of the individual state of health. Interestingly, few studies reported an overall improvement in subjective health in the wake of the COVID-19 pandemic [10,59,60]. The German LORA study showed a decrease in subjectively perceived stress and strain. The authors hypothesize that the effects of lockdowns may have led to a reduction in everyday stressors such as long commutes or a reduced workload [61]. Furthermore, due to the positive effects of the pandemic measures, some authors discuss a change in assessment behavior in times of health crises. Non-infected people may assess their health more positively in times of health crises than under everyday circumstances. Moreover, the challenge of the pandemic may have led to a more mindful approach to one’s health, which might be an explanation for the results obtained in our survey [60].

### 4.3. Attitudes towards WFH

Overall, a positive attitude towards WFH was observed. However, concerning the feelings of isolation, opinions of the respondents differed. Among all participants, the largest proportion of respondents stated that WFH promotes feelings of isolation, whereas those who WFH tended to disagree. Other studies rather show an increase in loneliness during the pandemic [48,62,63], e.g., based on the data from the “German Socio-Economic Panel”, the feeling of loneliness among people living in Germany has increased significantly during the crisis [48]. It is suspected that prolonged WFH and the resulting reduction in social interaction may have led to loneliness and isolation as well as depression [63]. Additionally, legal contact restriction measures during the pandemic may have led to an increase in perceived loneliness.

Most respondents who WFH stated that home-based working reduced stress. A similar result was observed concerning all our participants. Other studies confirm our findings in which WFH, especially at the beginning of the pandemic, was predominantly perceived positively and showed little impact on stress and well-being [64,65,66]. On the other hand, some studies have also shown the negative effects of WFH on perceived stress. In particular, parents of younger children showed increased signs of stress [67]. Other factors that might have a negative impact are an increase in sedentary work and thus a reduction in physical activity, as these factors are known to have a pain-relieving effect and can reduce stress [67]. We hypothesize that framework conditions such as workplace equipment, working time flexibility, corporate culture, or the parallel care of small children play a central role in stress, and WFH should continue to be monitored to record long-term effects.

Our study did not reveal a clear trend regarding the compatibility of childcare with WFH as most of the respondents abstained. In contrast, some studies tend to show a negative influence of childcare on stress while WFH [68,69]. For instance, the study conducted by Juncke et al., in which 750 companies throughout Germany and 1493 fathers and mothers with children under 15 were surveyed, 23% of parents stated that, in addition to work and household chores, life had become more stressful due to childcare and homeschooling [68].

Furthermore, we observed a clear desire among WFH employees to continue home-based work after the pandemic. This result is in line with the current literature [62,70,71]. Kunze et al. found that most respondents would prefer to WFH 2–3 days a week as it offers advantages, such as flexible working hours or more autonomy [62]. Interestingly, the enthusiasm for home-based work after the pandemic was less positive among both the whole cohort and those who were not WFH. This can possibly be explained for these groups by feared disadvantages like, for example, social isolation or communication issues.

In addition, most WFH employees stated that they were more productive at home. Other studies came to a similar conclusion [62,72]. This indicates that, despite the changed working conditions, employees and managers have worked together efficiently and that cooperation and working at a distance is possible. In contrast, there are studies reporting lower self-assessed productivity of employees [73,74]. Possible reasons for lower productivity could have been, e.g., insufficient technical knowledge/equipment, communication problems, less motivation, distraction at home, or poor leadership.

It was found that self-managed working time at home was under control for WFH participants. In contrast, few studies show a tendency toward more work hours when WFH [62,75]. Possible causes discussed include process-related restrictions or the removal of limits on working hours.

Regarding working hours, the compatibility of WFH and private life is also important. The results of our study showed no negative effect of WFH on the partnership whereby the agreement of both the whole cohort and the participants not WFH was less pronounced than among the participants who worked from home. This was in line with the current literature [52,61,76,77]. Kunze et al. showed that more than 70% of respondents rated the opportunity to combine work and private life at WFH as positive, regardless of whether or not there were minor children to look after [62]. Greater flexibility regarding working hours and place of work could be reasons for a better work–life balance. Conversely, a minority reported a lack of work–life balance and an increase in conflicts between family members [51].

In addition, a clear majority of our WFH participants were satisfied with their personal WFH situation. Our study results are in line with other current research findings [78,79]. For instance, in the survey conducted by the Bavarian Research Institute for Digital Transformation (bidt), the vast majority (87%) of respondents who WFH was satisfied or very satisfied with their WFH situation [71]. The main reason for this result may be the greater flexibility in terms of time and space.

### 4.4. Influence of Working from Home on the Psychological and Physical Strain

Overall, the attitudes towards WFH were rather positive in our survey, especially among those respondents who worked from home. Despite the high satisfaction of our cohort with working from home and many other positively rated characteristics such as the feeling of less stress, a good work–life balance, more productivity, or good control over working hours, we could not find any significant effects on either mental or physical well-being when comparing employees who worked from home with those who did not. Accordingly, we were unable to establish a connection between WFH and mental or physical well-being. Many other studies show that WFH can have both positive and negative effects on employees’ well-being [13,80]. The employee’s working environment in the home office appears to be particularly important in this regard. Influencing factors such as, e.g., caring for young children, little support from the company as well as less contact to colleagues, longer sitting times, or greater workload appear to be significant [20]. Our sample is characterized by a homogeneity of age with an average age of 50 years and at least a 10th grade school degree. These characteristics could be one explanation for a weaker influence of WFH on well-being. Moreover, the survey was conducted quite late in the course of the pandemic. The experience of the previous pandemic years may have led to habituation effects and the adoption of new strategies to deal with this crisis.

### 4.5. Strengths and Limitations

The major strength of the study is the cohort of SLS followed up for more than 35 years which qualifies it as one of the largest longitudinal studies in the German-speaking area. This study design also has several advantages. Firstly, the data were collected since 1987 at almost annual intervals from selected individuals, thus appropriate statistical tests can be used to analyze changes over time for the group as a whole or for specific individuals (such as WFH respondents in our case). Secondly, long-term evaluations make it possible to assess the relationship between risk factors such as the coronavirus pandemic and the development of diseases or long-term consequences. Thirdly, since the cohort studied is fixed, changes over time are only slightly influenced by cohort differences [81]. On the other hand, the participants included in the SLS are an age-homogeneous group. Consequently, it is not possible to generalize the results to all age groups, which is a limitation of the study. In addition, all study participants were born in the federal state of Saxony in East Germany (former GDR) and most of them also resided there at the time of the survey. The responses of participants who were living in West Germany or abroad at the time of the survey cannot be used without restriction as comparative values, as this group consists exclusively of people who immigrated from the west. These may differ in their basic life and value concepts from people born and raised in West Germany. Although a comparison with data from a corresponding West German study would be feasible, there is a lack of an adequate, analogous cohort. In summary, the results can be generalized to the population of eastern Germany, but not to the German population as a whole. In terms of educational level, the participants in the SLS have an above-average level of education due to the study design (all persons have at least a 10th grade degree; 38.7% have a general university entrance qualification or higher). Possible biases, especially those caused by factors associated with education (e.g., income), cannot be ruled out [24]. In addition, a longitudinal study also poses challenges, such as incomplete and interrupted follow-up of individuals with loss of follow-up over time. Thus, there could be a loss of representativeness of the dynamic sample [82]. The response rate decreased over the years. Of the *N* = 587 people who continued to participate in 1989, only 54.3% still took part in the study in 2022, 33 years later. The reasons for the non-participation of individual participants are not always known and, therefore, cannot always be determined. Moreover, the sample is limited because participants did not consistently take part in all surveys [24].

Short scales from established survey instruments were used in our study to assess mental and physical strain. On the one hand, meaningful results can be obtained with a reduced number of precise questions, and the compactness of the short scale also supports participant compliance. On the other hand, although the short scales are validated, they are still very short screening instruments. Complex issues are assessed with the help of a few questions, which can lead to a bias of answers. Additionally, the WFH questionnaire was newly constructed from selected questions of the “Homeoffice Barometer” of the research institute gfs.bern, and, therefore, it was not yet psychometrically validated. In addition, the longitudinal study design limited the number of socio-demographic characteristics examined. Different levels of education, migration backgrounds, or different age groups were not included, which may have had an influence on WFH as well as on physical and psychological strain. Moreover, we examined exclusively the aspect of WFH or not WFH. Other influencing factors, e.g., the equipment at home, the corporate culture, or communication within the team, should be investigated in additional surveys. In light of the restrictions, further follow-up studies on the mental and physical stress of employees WFH during the COVID-19 pandemic and beyond are needed.

## 5. Conclusions

In summary, we were able to show that the coronavirus pandemic had a negative influence on psychological and physical strain. Therefore, companies should pay particular attention to the mental and physical health of their employees. The health of employees is crucial to their well-being and performance in the workplace. A healthy working environment, ergonomic office furniture, and the promotion of a healthy lifestyle (e.g., sports exercises at work, reduction in sitting hours, healthy cafeteria meals) could help to minimize physical health risks. To cope with work-related mental stress, mental health problems and maintain a healthy work–life balance, companies should provide preventive and supportive measures. This may include the provision of resources such as counseling services, or flexible working hours.

Moreover, we observed a very popular willingness to WFH among surveyed participants. Although the associated negative effects, such as increased feelings of being alone, were observed, they contrasted with a stable subjective health perception, high satisfaction, less stress, as well as good productivity and work–life balance. Interestingly, WFH does not appear to have an impact on health during the pandemic. Nevertheless, companies should take advantage of the pandemic-driven shift in workplace design towards WFH. Based on our findings, this option makes the workplace more attractive, as it offers greater flexibility, a good work–life balance, and employee satisfaction. Given the growing popularity of home-based work, further studies are needed to uncover a possible link between mental and physical strain in pandemic-free years.

## Figures and Tables

**Table 1 healthcare-12-00789-t001:** Sociodemographic structure of the SLS between 2017 and 2022.

Sociodemographic Characteristics	Wave 30 (2017), *N* = 314	Wave 31 (2019), *N* = 323	Wave 32 (2021), *N* = 321	Wave 33 (2022), *N* = 319
**Gender**				
Male, *N* (%)	143 (45.5%)	143 (44.3%)	149 (46.4%)	145 (45.5%)
Female, *N* (%)	171 (54.5%)	180 (55.7%)	172 (53.6%)	174 (54.5%)
**Age**				
Median (Min; Max)	45.00 (44; 52)	47.00 (46; 54)	48.00 (47; 455)	50.00 (49; 57)
**Partnership**				
Yes, *N* (%)	251 (81.2%)	254 (81.4%)	252 (80.8%)	252 (81.8%)
No, *N* (%)	58 (18.8%)	58 (18.6%)	60 (19.2%)	56 (18.2%)
**Own children living in the household**				
No child, *N* (%)	26 (10.5%)	39 (15.2%)	54 (21.5%)	75 (29.8%)
One child, *N* (%)	102 (42.3%)	123 (48.0%)	110 (43.8%)	105 (41.7%)
More than one child, *N* (%)	113 (47.2%)	94 (63.8%)	87 (34.7%)	72 (28.5%)

*N* (Number of participants); Min (Minimum); Max (Maximum).

**Table 2 healthcare-12-00789-t002:** Course of the psychological and the physical strain scores.

W30 (2017)		W31 (2019)		W32 (2021)		W33 (2022)		Wilcoxon Test		
*N*	Median (IQR 25th; 75th)	*N*	Median (IQR 25th; 75th)	*N*	Median (IQR 25th; 75th)	*N*	Median (IQR 25th; 75th)	Group Comparisons	Test Statistics	*p*-Value
**D-Score**										
312	0.00 (0.0; 1.0)	320	0.00 (0.0; 1.0)	320	0.00 (0.0; 1.0)	319	0.00 (0.0; 2.0)	W31 vs. W32	Z = −0.555	*p* = 0.579
								W31 vs. W33	Z = −3.939	***p* < 0.001 **
								W32 vs. W33	Z = −3.690	***p* < 0.001 **
**PHQ-2**										
305	2.00 (2.0; 3.0)	319	2.00 (2.0; 4.0)	-	-	319	3.00 (2.0; 4.0)	W30 * vs. W31	Z = −0.378	*p* = 0.705
								W30 * vs. W33	Z = −6.545	***p* < 0.001 **
								W31 vs. W33	Z = −6.113	***p* < 0.001 **
**GAD-2**										
311	3.00 (2.0; 4.0)	316	3.00 (2.0; 4.0)	-	-	317	3.00 (2.0; 4.0)	W30 * vs. W31	Z = −1.070	*p* = 0.284
								W30 * vs. W33	Z = −5.170	***p* < 0.001 **
								W31 vs. W33	Z = −5.419	***p* < 0.001 **
**PHQ-4**										
305	5.00 (4.0; 7.0)	313	5.00 (4.0; 7.0)	-	-	317	6.00 (4.0; 8.0)	W30 * vs. W31	Z = −0.625	*p* = 0.532
								W30 * vs. W33	Z = −6.262	***p* < 0.001 **
								W31 vs. W33	Z = −6.047	***p* < 0.001 **
**Corona Anxiety Scale**										
-	-	-	-	315	17.00 (13.0; 21.0)	318	14.00 (11.0; 18.0)	W32 vs. W33	Z = −7.630	***p* < 0.001 **
**LS-S**										
311	6.00 (4.0; 7.0)	-	-	-	-	319	6.00 (4.0; 7.0)	W30 ** vs. W33	Z = −0.073	*p* = 0.942
**L-1**										
312	9.00 (8.0; 10.0)	323	9.00 (8.0; 10.0)	318	9.00 (8.0; 10.0)	317	9.00 (8.0; 10.0)	W31 vs. W32	Z = −0.257	*p* = 0.797
								W31 vs. W33	Z = −0.582	*p* = 0.561
								W32 vs. W33	Z = −0.693	*p* = 0.488
**G-Score**										
311	3.00 (1.0; 5.0)	323	3.00 (2.0; 6.0)	320	3.00 (1.0; 5.0)	319	3.00 (2.0; 5.0)	W31 vs. W32	Z = −2.076	***p* = 0.038 **
								W31 vs. W33	Z = −0.270	*p* = 0.787
								W32 vs. W33	Z = −1.647	*p* = 0.100
**Individual Health Status**										
311	2.00 (2.0; 3.0)	320	2.00 (2.0; 3.0)	320	2.00 (2.0; 3.0)	319	2.00 (2.0; 3.0)	W31 vs. W32	Z = −0.403	*p* = 0.687
								W31 vs. W33	Z = −0.983	*p* = 0.326
								W32 vs. W33	Z = −0.492	*p* = 0.623
**SSS-8**										
311	5.00 (2.0; 9.0)	-	-	-	-	318	6.00 (3.0; 11.0)	W30 ** vs. W33	Z = −3.941	***p* < 0.001 **

IQR 25th, 75th (interquartile range (25th and 75th percentile)); W (wave); *N* (Number of participants); Numbers in bold indicate significant values, level of significance α = 0.05; * W30 selected because PHQ-2 and GAD-2 were not surveyed in wave 32; ** W30 selected because no values are available from waves W31 and W32.

**Table 3 healthcare-12-00789-t003:** Attitude towards WFH, wave 33.

**Questions**	**Participants (Whole Cohort),** ***N* (%)**			**Participants WFH,** ***N* (%)**			**Participants Not WFH,** ***N* (%)**			**Information on WFH Not Applicable**	**Chi^2^-Test** **(df = 4)**
	**Agreement**	**Do Not Know/No Answer**	**Disagreement**	**Agreement**	**Do Not Know/No Answer**	**Disagreement**	**Agreement**	**Do Not Know/No Answer**	**Disagreement**		**WFH vs. Not WFH**
WFH promotes the feeling of being alone	118 (38.3%)	115 (37.3%)	75 (24.3%)	32 (38.1%)	12 (14.3%)	40 (47.6%)	83 (38.8%)	97 (45.3%)	34 (15.9%)	10	TS = 4.674
											***p* < 0.001**
WFH reduces stress at work	129 (41.9%)	126 (40.9%)	53 (17.2%)	44 (52.4%)	15 (17.9%)	25 (29.7%)	81 (37.9%)	106 (49.5%)	27 (12.6%)	10	TS = 31.262
											***p* < 0.001**
WFH is not compatible with childcare	68 (22.2%)	153 (49.8%)	86 (28.0%)	17 (20.2%)	33 (39.3%)	34 (40.5%)	48 (22.5%)	113 (53%)	52 (24.4%)	10	TS = 9.334
											*p* = 0.053
I would also like to work (partly) from home after the pandemic	90 (29.6%)	119 (39.1%)	95 (31.3%)	68 (81.0%)	5 (6.0%)	11 (13.1%)	22 (10.3%)	106 (50%)	84 (39.6%)	8	TS = 158.25
											***p* < 0.001**
I can work more productively at home	70 (23.4%)	140 (46.8%)	89 (29.8%)	49 (59.0%)	8 (9.6%)	26 (31.3%)	20 (9.5%)	128 (61%)	62 (29.5%)	6	TS = 10.354
											***p* < 0.001**
I am not in control of my working time at home	43 (14.5%)	138 (46.5%)	116 (39.0%)	19 (22.9%)	9 (10.8%)	55 (66.2%)	23 (11.1%)	125 (60.1%)	60 (28.8%)	6	TS = 66.809
											***p* < 0.001**
My family/partnership suffers from WFH	15 (5.1%)	155 (52.5%)	125 (42.4%)	3 (3.6%)	9 (10.8%)	71 (85.5%)	11 (5.3%)	141 (68.4%)	54 (26.2%)	6	TS = 91.359
											***p* < 0.001**
	**Participants (Whole Cohort),** ***N* (%)**			**Participants WFH,** ***N* (%)**			**Participants Not WFH,** ***N* (%)**			**Information on WFH Not Applicable**	**Chi^2^-test** **(df = 4)**
	**Satisfaction**	**Not Assessable**	**Dissatisfaction**	**Satisfaction**	**Not Assessable**	**Dis-Satisfaction**	**Satisfaction**	**Not Assessable**	**Dis-Satisfaction**		**Satisfaction**
How satisfied are you with your personal WFH situation?	95 (32.3%)	184 (62.6%)	15 (5.2%)	72 (84.7%)	9 (10.6%)	4 (4.7%)	23 (11.3%)	169 (83.3%)	11 (5.4%)	6	TS = 152.018
											***p* < 0.001**

WFH (working from home); *N* (number of participants); TS (test statistic); Numbers in bold indicate significant values, level of significance α = 0.05.

**Table 4 healthcare-12-00789-t004:** Psychological and physical health of employees who WFH (*N* = 86, 28.1%) compared to those working on-site (*N* = 221, 71.9%).

Scores	Median (IQR 25th; 75th)	Wilcoxon Test	*p*-Value
**D-Score**			
WFH	0.00 (0.0; 1.0)	Z = −1.307	*p* = 0.191
Not WFH	0.00 (0.0; 2.0)		
**PHQ-4**			
WFH	7.00 (4.0; 8.0)	Z = 0.873	*p* = 0.350
Not WFH	6.00 (4.0; 8.0)		
**PHQ-2**			
WFH	3.00 (2.0; 4.0)	Z = −1.104	*p* = 0.269
Not WFH	3.00 (2.0; 4.0)		
**GAD-2**			
WFH	3.00 (2.0; 4.0)	Z = −0.590	*p* = 0.555
Not WFH	3.00 (2.0; 4.0)		
**Corona Anxiety Scale**			
WFH	14.5 (12.0; 17.0)	Z = −0.310	*p* = 0.756
Not WFH	14.00 (11.0; 19.0)		
**LS-S**			
WFH	6.00 (4.0; 7.0)	Z = −0.610	*p* = 0.542
Not WFH	6.00 (4.0; 7.0)		
**L-1**			
WFH	9.00 (8.0; 10.0)	Z = −0.580	*p* = 0.562
Not WFH	9.00 (8.0; 10.0)		
**G-Score**			
WFH	3.00 (2.0; 5.0)	Z = −0467	*p* = 0.640
Not WFH	4.00 (2.0; 6.0)		
**Individual Health Status**			
WFH	2.00 (2.0; 3.0)	Z = −0.769	*p* = 0.442
Not WFH	2.00 (2.0; 3.0)		
**SSS-8**			
WFH	5.00 (3.0; 11.0)	Z = −0.489	*p* = 0.625
Not WFH	6.00 (3.0; 11.0)		

M (mean); SD (standard deviation); W (wave); *N* (number); WFH (working from home).

## Data Availability

The analysis is based on the Saxon Longitudinal Study. The data as well as all questionnaires with their individual items of the Saxon Longitudinal Study are archived at the Leibniz Institute for the Social Sciences (gesis) and can be obtained for research purposes at https://search.gesis.org/research_data/ZA6249 (accessed on 13 March 2024), https://search.gesis.org/research_data/ZA7841 (accessed on 13 March 2024), https://search.gesis.org/research_data/ZA7842 (accessed on 13 March 2024), https://search.gesis.org/research_data/ZA6248 (accessed on 13 March 2024).

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
