# Peer review of "Impact of Home-Based Work during the COVID-19 Pandemic on Mental and Physical Health in a German Population-Based Sample"

_healthcare, 2024, doi:10.3390/healthcare12070789_

Round 1

Reviewer 1 Report

Comments and Suggestions for Authors

Dear authors, 

Thank you for submitting your research! The results are interesting and the conclusions are important. However I have several comments: 

Table 1: you reported the age and the number of children as mean and SD but did you check the distribution of the variables for normality? If yes, please specify the test within Statistical analysis section. For the number of children probably a proportion of those living with 1 child would be better. The age of your cohort is changing each year so probably it would be better to present the year of birth as median, min and max. 

Page 3, 2.3. Measurement: Your sentence is too long. You can omit "such as gender, age, partnership, and own children"

2.3.1. You have to report the internal consistency measures for all study years included in the article. 

You used a lot of questionnaires. Ethics require to reveal if any incentives are given to the respondents. 

2.4. Statistical analysis

Please add the name of the non-parametric test you used for the independent samples comparison (Table 4). 

Results:

Table 2 show means and SD. You claimed the usage of the non-parametric Wilcoxon test. So it would be better to show medians and interquartile range (25th and 75th percentile) instead of the means and SD. In addition, the text "mean difference" should be replaced with "average difference" since you do not compare the mean values. The test itself compares mean ranks. 

3.2. Physical strain: a dash is missing in the brackets (2017-2022)

line 260 - the German word "albeit" appears

Table 3: it does not make sense to compare a part of the sample to the entire sample. The only comparison that is needed is between WFH and not WFH. The numbers and proportions for the entire sample should remain in the table but please carefully check if they equals when adding WFH and not WFH numbers. 

The last part of the table that appears on page 10 seems to have a problem with its title row for the last 3 columns. 

3.4.

Here you mentioned a non-parametric test, so please add its name within Statistical analysis section as I previously asked. It could be Mann-Whitney U test or other applicable. 

Table 4: the same as table 2: instead of mean and SD, median and interquartile range should be reported. 

Comments on the Quality of English Language

Page 7, line 260, the word "albeit" is German. 

COVID should be entirely capitalized (on page 5, line 217 it is not)

Some words start with a capital letter as it is required in German language but not in English. For exapmle page 12, line 39 "In Particular". 

Author Response

Dear reviewer, 

We uploaded a clean version of the revised paper (word-version), and a separate marked-up PDF-version (with visible track changes).

Sincerely, 

the authors 

Reviewer 2 Report

Comments and Suggestions for Authors

This study aimed to investigate the association between home-based work during the COVID-19 pandemic and several psychological conditions.

I have the following comments:

1: There are several abbreviations in the abstract (The D- 23 score, PHQ-4, LS-S, and L-1 were collected to assess mental health). The authors should have spelled them out and described them in brief. 

2: In Table 3, the sum of WFH and non-WFH is lower than the overall sample.

3: One of the main limitations of this study is that several sociodemographic and occupational factors could affect the association between home-based work and psychological indices, yet these factors were not included in the analysis.

Author Response

(The authors gave the same response as above.)

Reviewer 3 Report

Comments and Suggestions for Authors

The article I received for review examines the impact of working from home on mental and physical health in a German population-based sample in the context of the COVID-19 pandemic. The topic addressed in the article is highly relevant, especially in view of the global health crisis that the pandemic caused. The authors attempt to understand how remote working has affected the wellbeing and fitness of workers.

The thematic value of this article is immense, especially in the context of the changes that are taking place in the way work is done in times of pandemic. The analysis of the impact of remote working on mental and physical health is extremely relevant for both employers and employees, as well as for professionals working in public health and organisational psychology.

However, despite the interesting topic and relevance of the issue, the article has some shortcomings. The introduction inadequately highlights the research gap and does not sufficiently explain what new insights the presented study brings to existing knowledge. There is also a lack of clearly stated research questions or hypotheses to guide the analysis of the results.

The 'Material and Methods' section lacks a detailed description of the selection of the research sample and the different stages of the study. There is also insufficient justification for specifically choosing Saxony in Germany as the research area. In addition, it would have been useful to present the research tools in an appendix with their individual items and factor loadings.

The authors could also have described in more detail the implications for managerial practice related to the use of remote working. The presentation of practical tips for employers could significantly enrich the article.

Author Response

(The authors gave the same response as above.)
